# SIMPLE MECHANISMS FOR REPRESENTING, INDEXING AND MANIPULATING CONCEPTS

## ABSTRACT

Deep networks typically learn concepts via classifiers, which involves setting up a model and training it via gradient descent to fit the concept-labeled data. We will argue instead that learning a concept could be done by looking at its moment statistics matrix to generate a concrete representation or signature of that concept. These signatures can be used to discover structure across the set of concepts and could recursively produce higher-level concepts by learning this structure from those signatures. Concepts can be 'intersected' to find a common theme in a number of related concepts. This process could be used to keep a dictionary of concepts so that inputs could correctly identify and be routed to the set of concepts involved in the (latent) generation of the input.

## 1 INTRODUCTION

Deep networks typically learn concepts via classifiers Qi et al. (2017); Allen-Zhu & Li (2020); Längkvist et al. (2014); Zhong et al. (2016); Jing & Tian (2020); Guo (2021), which involves setting up a model and training it via gradient descent to fit the concept-labeled data. However, this does not provide a general understanding of what a concept should be and how they may be discovered and represented. In this work, we attempt to provide a formalism of concepts that addresses these issues. Intuitively, we understand a concept to be a collection of objects with a shared set of properties. In the paper, we consider such properties as defined by a polynomial manifold, i.e., the zero set of polynomial equations.

While there has been work on learning manifolds Izenman (2012); Caterini et al. (2021); Bashiri et al. (2018); Pedronette et al. (2018); Wang et al. (2018); Lin & Zha (2008); Brehmer & Cranmer (2020); Brosch et al. (2013); Han et al. (2022); Cayton et al. (2008); Ma & Fu (2011); Zhu et al. (2018); Lunga et al. (2013); Talwalkar et al. (2008), our goal here is to find simple ways to automatically group points into collections of manifolds and obtain simple representations of these manifolds. We show that by simply taking the moment statistics corresponding to such points and looking at the null-space of these statistics (arranged as a matrix), we can identify the manifold. In other words, these elementary statistics can serve as a *signature* of the concept. Thus, we are learning a concept by tracking its elementary statistics. This signature can serve as a classifier and as a representation or signature of that concept.

Importantly, these signatures can be used to discover structure across the set of concepts and could recursively produce higher-level concepts by learning this structure from those signatures. Concepts may be "intersected" to find a common theme across a number of related concepts. For a simple illustration of this abstraction, see Figure 1.

We also map the process of concept discovery to the transformer architecture Vaswani et al. (2017) at an intuitive level. We view a generalized transformer architecture with attention blocks that are used to group inputs that are part of the same concept. The MLP layers could then be used to compute the moment statistics and the corresponding null-space leading to the signature of manifolds. We also augment each layer with a dictionary of already discovered concept signatures (viewed as a memory table of concepts). These tables can be looked up to identify known concepts that are present in a given input. This process is repeated iteratively producing a hierarchy of concepts.

While there have been many interesting experimental works showing that the feedforward networks encode concept information Geva et al. (2022), visualizing the feed-forward network as a key-value memory unit Geva et al. (2021) or keeping the explicit memory component to store the knowledge Lewis et al. (2020); Wu et al. (2022), they fail to explain how the concepts are discovered and how the lower level concepts are combined to obtain the higher-level concepts.

## 1.1 OUR CONTRIBUTION

Our main results can be summarized as follows:

**Proposition 1.1.** (Informal summary of results) Given a collection of concepts where each concept corresponds to a (latent) low dimensional, low curvature manifold, there is a simple method to compute concept signatures such that

1 The concept signature is obtained by taking moment statistics of a few points within the manifold, followed by applying a power iteration-like transform that results in a signature that is invariant across different regions in the manifold. That is, regardless of which region in the manifold the points may be distributed from, the resulting concept signature will be the same as long as the distribution has a "well-conditioned" structure (Proposition 2.1).

2 A simple attention mechanism may be used to identify a set of points from the same manifold (Proposition 3.1). This simply involves looking at the inner product of the two points after a simple kernel transform. Membership in a manifold is also a simple inner product check between the concept signature and this kernel transform representation of the point (Theorem 2.1).

3 If there is structure across a collection of concepts, then this structure is also present in the concept signatures. For example, a concept may be the intersection of two other concepts; that is the manifold is the intersection of the two corresponding manifolds; then this structure can be easily inferred from the concept signatures. If several similar concepts share a common underlying (latent) concept, the common concept signature can be easily obtained from a few of those related concepts (Proposition 3.2).

  If there is a dictionary of atomic concepts and all concepts are unions of these atomic concepts, then the signature of the atomic concepts can be inferred (Lemma 3.2).

4 Higher-level concepts may be obtained when concept signatures of lower level concepts lie in a manifold and as long as the dimension of manifold of each lower–level concept is constant and can be parameterized by constant degree polynomial, the signature sizes of the higher level concept will be constant up the hierarchy.

  For example, a single rectangle is a concept that includes the concept of the four line segments. Each line segment has a concept signature and these four signatures line a subspace. The general concept of a circle/rectangle is obtained by taking signatures of several circles/rectangles that all lie on a single abstract manifold and computing the concept signature of that manifold (see Lemma 4.1 for circle).

  The video of an object moving in a specific motion pattern – for example, a car turning right will result in a concept signature that will be related to that of any other object moving in the same pattern. By intersecting the two concept signatures we obtain the concept of "moving right" (see Lemma 4.4).

  Similarly rotated copies of an image result in concept signatures that lie in a manifold. These manifolds corresponding to rotations of different images share a common rotation manifold that captures the concept of rotating any image. The concept signature of rotation can be used to take the signature of an image and convert it into a rotation invariant signature which is the signature of the manifold that includes all rotations of that object (Lemma 4.3).

5 There are simple instantiations of a transformer network with mostly random set of parameters and a few learned projections that can be used to group together and compute concept signatures of latent manifolds present in the input data (see Section 6).

Our analysis points towards an architecture that is very similar to the transformer but in addition also has a dictionary of concepts signatures at each layer that are produced over time in a system that is receiving a continuous stream of inputs (similar to the experimental works that keep an explicit memory unit to store the knowledge Lewis et al. (2020); Wu et al. (2022)); this set of signatures is looked up as new inputs arrive into the system.

## 2 CONCEPTS AS SUBSPACE SIGNATURES

In this section, we define our main notion of a *signature* for a set of points that is meant to capture the notion of concepts. We then show that when the underlying points lie on an *algebraic manifold* (see definition below), the signature we define satisfies the conceptual requirements outlined in the introduction.

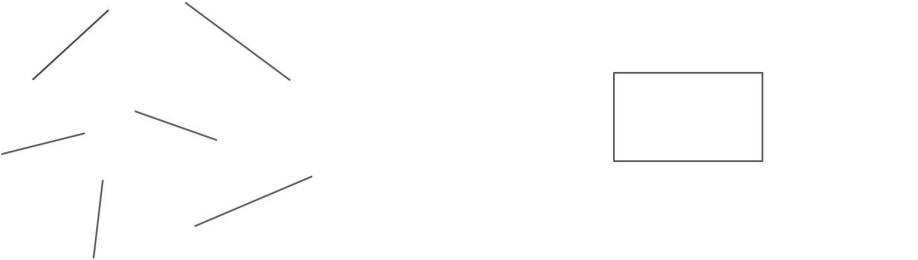

(a) Several lines on a 2-d plane, each with a separate signature.

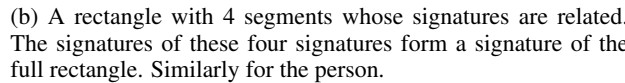

(b) A rectangle with 4 segments whose signatures are related. The signatures of these four signatures form a signature of the full rectangle. Similarly for the person.

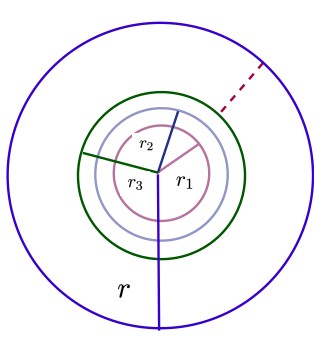

(c) Several concentric circles of varying radius: the signatures of the individual circles all lie on a manifold that is the signature of the concept of a circle.

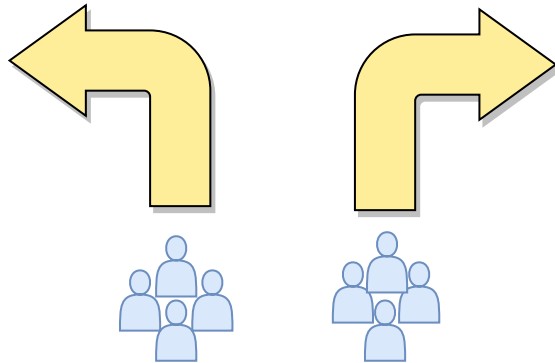

(d) Two groups each traveling along a different trajectory. The trajectory of each object produces a signature; all objects in one group have related signatures that lie in a manifold.

Figure 1: Examples and properties of the signatures.

We will use the following notational conventions: For two tensors $A, B$ of the same dimension, we use $A.B$ to denote $\langle A, B \rangle \in \mathbb{R}$. For any vector $x$, $x^{\cdot l}$ denotes the $l$'th tensor power of $x$.

**Definition 2.1.** Given a feature mapping $\phi : \mathbb{R}^d \to \mathbb{R}^m$ and a collection of points $x_1, x_2, \cdots, x_N \in \mathbb{R}^d$, we define the kernel signature of $X = (x_1, \cdots, x_N)$ as:

$$M = M(X) = \frac{1}{N} \sum_{i \in [N]} \phi(x_i)\phi(x_i)^\top \in \mathbb{R}^{m \times m}$$

For a single point, we also define $S(x) = \phi(x)\phi(x)^\top$.

Given $U\Sigma U^\top$ be the Singular Value Decomposition (SVD) of $M$, where $\Sigma_1 \geq \Sigma_2 \geq \Sigma_k > 0, \Sigma_{k+1}, \cdots, \Sigma_m = 0$, we define the null-space signature as $T(X) = U_{k:m}U_{k:m}^\top$. We define the $\epsilon$-approximate null-space signature as $T_\epsilon(X) = U_{k':m}U_{k':m}^\top$, where $\Sigma_{k'} \leq \epsilon$ and $\Sigma_{k'-1} > \epsilon$.

**Definition 2.2** (Signature). A signature of a set of points is defined as $(M(X), T(X), T_\epsilon(X))$ for some prescribed $\epsilon$.[1]

**Definition 2.3** (Algebraic Manifold). A set $\mathcal{M} \subseteq \mathbb{R}^d$ is an algebraic manifold if $\mathcal{M} = \{x : P(x) = 0\}$ where $P : \mathbb{R}^d \to \mathbb{R}^{d-k}$ is a set of $d - k$ polynomial maps. The manifold in this case is $k$-dimensional and we say the degree of the manifold is the max degree of the polynomials $P_i$.

Throughout, we adopt the convention that the defining polynomial equations satisfy the normalization $E_{y \in \mathbb{S}^{d-1}}[P_i(y)^2] = 1$ for $i \in [d - k]$ where the notation $\mathbb{S}^{d-1}$ denotes the unit-sphere in $d$-dimensions.

---

[1] We keep $T_\epsilon(X)$ so in the later section, we will discuss if $X$ only approximately lies on the manifold $\mathcal{M}$, but not exactly.

Given a $k$-dimensional manifold as above, if it exists, we call $G : \mathbb{R}^k \to \mathbb{R}^d$ a generative representation of $\mathcal{M}$ if the following holds: for all $x \in \mathbb{R}^d$, $x \in \mathcal{M}$ if and only if there exists $z \in \mathbb{R}^k$ such that $x = G(z)$.

More generally an *analytic manifold* is obtained when the functions $G$ are analytic (possibly in a bounded region).

We will look at manifolds where the dimensionality $k$ and the degree are small constants – as a simple example a circle is a 1-d manifold of degree 2. We will argue later how rich concepts can be formed by combining several such simple concepts hierarchically.

**Definition 2.4** (Non-degenerate distribution)**.** We will say that a distribution of points from manifold is non-degenerate if for any subset of features in $\phi(x)$ that are linearly independent over the entire manifold are also linearly independent over the support of the distribution. For example, if $\phi$ a polynomial kernel, and the manifold is analytic, a distribution supported on a (even a small) ball on the manifold is non-degenerate (this is because if a polynomial is identically zero within a ball then it is identically zero over the entire analytic manifold)

**Proposition 2.1.** [Moments null space is zero polynomials on the manifold] Let $X = \{x_1, \ldots, x_n\}$ be random points drawn from a $k$-dimensional algebraic manifold $\mathcal{M}$ of degree $\ell$.

Under suitable non-degeneracy conditions on $\mathcal{M}$, for sufficiently large $n$, if we compute the signature $T(X)$ with $\phi$ being the degree $\ell$-polynomial feature mapping (i.e., $\phi(x)$ contains all monomials of $x$ of up to degree $\ell$), then the signature $T(X)$ as in Definition 2.1 uniquely identifies the manifold $\mathcal{M}$. Membership check: A point $x \in \mathcal{M}$ iff $S(x).T(X) = 0$, otherwise $S(x).T(X) > 0$.

*Proof.* Let $p : \mathbb{R}^n \to \mathbb{R}$ be a polynomial that vanishes on the manifold $\mathcal{M}$. Then, for any point $x$, $p(x) = \langle w, \phi(x) \rangle$ (here we view $w$ as the coefficient vector of the polynomial $p(x)$). Thus, if $p(x) = 0$ for all $x \in \mathcal{M}$, then $w$ is in the null-space of $M(X)$ as $w^T \phi(x_i)\phi(x_i)^T w = 0$ for all $x_i$.

For large enough $n$, assuming $\mathcal{M}$ is non-degenerate, any polynomial that is non-zero on a ball of the manifold will not be in the null-space of $M(X)$ (with high probability). Thus, a point $x$ is in the null-space if and only if $\phi(x)^T T(X)\phi(x) = 0$, i.e., $S(x).T(X)$ is 0. $\square$

For constant dimensional manifolds with a generative representation with constant degree polynomials, the sample-size required in the statement above can be reduced to a constant size (independent of the ambient dimension). The signature size can also be reduced to a constant size.

**Proposition 2.2.** [Invariant concept signatures] Suppose $X$ lies in a $k$-dimensional polynomial manifold with a degree $r$ generative representation. Suppose $\phi(x)$ is a degree $s$ monomial feature mapping, then $T(X)$ is uniquely defined by the signature as long as $X$ is non-degenerate and $N = \Omega\left(\binom{k^r+s}{s}\right)$.

*Proof.* We can write all the monomials of $x$ as a vector, whose coordinates are the coefficients in terms of $t \in \mathbb{R}^k$, where $t$ are the generative variables. We know that for degree $q$ monomials, there are at least $\binom{d}{q}$ such terms. On the other hand, we can write the degree $rq$ polynomials $p$ of $t$ as a vector in dimension at most $k^{rq}$. Taking $s = 1$, we know that the manifold lies in a $k^r$ dimensional subspace $U$.

We now need to prove that $T(X)$ is uniquely defined. To see this, let us consider a $w$ such that $\langle w, \phi(x_i) \rangle = 0$ for all $i \in [N]$. We need to show that $\langle w, \phi(x) \rangle = 0$ for all $x \in Y$. Note that by our previous conclusion,

$$\phi(x) = \phi(UU^\top x)$$

Therefore we can restrict ourself to polynomials $g(z) = \phi(Uz)$, where $z \in \mathbb{R}^{k^r}$.

Now, for each polynomial $g(z)$, we know that $g(U^\top p(t))$ is of degree $rs$. Therefore, for non-degenerate $t_1, \cdots, t_N$ where $N = \Omega(rs)$, $g(U^\top p(t_i)) = 0, \forall i \in [N] \implies g(U^\top p(t)) = 0$. Note that the VC dimension of $g$ is at most $\binom{k^r+s}{s}$ since $g$ is a polynomial of degree at most $s$ over input dimension $k^r$, so we know that the sample complexity is $O\left(\binom{k^r+s}{s}\right)$. $\square$

Now we want to quantitatively show that the signature can be used to do a membership check if the points lie in a manifold, thus the signature of those points will precisely describe the manifold.

We also show that we can reduce the sample complexity to $r^{\tilde{O}(k^2)}$ by using a random projection to a lower-dimensional space. The following follows easily from Theorem 1.6 in Clarkson (2008) (the statement there is about a $k$-dimensional manifold; we can view adding an extra point as a $(k+1)$-dimensional manifold).

**Lemma 2.1** (Reducing sample complexity using projection, Clarkson (2008))**.** Let $\mathcal{M} \subseteq \mathbb{R}^d$ be a connected, compact, orientable, differentiable manifold M with dimension $k$, constant curvature and bounded diameter. Then, for any $\epsilon, \delta > 0$ and a point $x, \mathbb{R}^d$, if $A : \mathbb{R}^d \to \mathbb{R}^m$ is a random linear map for $m = O((k + \log(1/\delta))/\epsilon^2)$, with probability at least $1 - \delta$, the following hold:

- For all $y, z \in \mathcal{M}$, $\|Ay - z\|_2 = (1 \pm \epsilon)\|y - z\|_2$.

- For all $y \in \mathcal{M}$, $\|Ax - Ay\| = (1 \pm \epsilon)\|x - y\|_2$.

We use the projection lemma to obtain the following.

**Theorem 2.1.** [$k$-dim manifold generated by degree $r$ polynomials] A $k$-dim manifold with generative representation $G$ that are polynomials of degree $r$, can be represented by a signature $T$ of size $r^{\tilde{O}(k^2)}$ (independent of ambient dimension $d$). Such a signature can be computed by projecting the points to a lower dimensional space and computing the signature as before. This signature can be used to test the membership of a point $x$ with high probability by checking the value of $S(x).T$: $S(x).T = 0$ if $x$ is on the manifold. In addition, for any point $y$ on the manifold, if we choose $z \in \{x : \|x - y\|_2 = \Delta\}$ at random, then $S(z).T \geq f_{k,r}(\Delta)$, where $f$ is an increasing function in $\Delta$ in a bounded region.

*Proof.* We first note that if there is a generative representation of a $k$-dimensional manifold by polynomials of degree $r$, then the degree of the manifold is at most $\ell = k^r$ - see Theorem 5.1.

Note that any manifold with generative representation of degree $r$ continues to have a generative representation of same degree after a projection.

Thus, if we use a random projection to project the space to $m = O((k + \log(1/\delta))/\epsilon^2)$, then by Lemma 2.1, all distances on the manifold and the test point $x$ are preserved with probability at least $1 - \delta$.

Let us work in the projected space of dimension $m$. Now, the number of monomials involved in the monomial map is $\binom{m+\ell}{\ell}$ which is at most $(m + \ell)^m$. Since $\ell$ is at most $r^k$, the dimension of the monomial map is $r^{km}$. Now, note that to approximate $T(X)$, we need a spectral approximation to $M(X)$. Recall that for any distribution on vectors in dimension $D$ with bounded covariance, the number of samples needed for the empirical covariance matrix to approximate the true covariance matrix is $\tilde{O}(D)$. By applying this to $\phi(x)$ (in the projected space), the number of samples needed for $M(X)$ to approximate the true moment matrix is $\tilde{O}(r^{km})$. Thus, the size of the signature is $\tilde{O}(r^{km}) = r^{\tilde{O}(k^2)}$.

Finally, for the last part, consider the polynomial $Q(x) = S(x).T$ (in the projected space). By our assumptions, the sum-of-squares of the coefficients of $Q$ is exactly the dimension of the null-space of $T$. The final part of the lemma now follows by Lemma C.4. $\qquad\square$

# 3 LEARNING ARCHITECTURE AUGMENTED WITH TABLE OF CONCEPT SIGNATURES AT EACH LAYER

We envision an architecture that is composed of mainly two modules - concept discovery module and concept storage module. The concept discovery module computes the null space signature on the current input and is composed of the standard transformer architecture whereas the concept storage module stores a dictionary of signatures $T_{\ell,1}, T_{\ell,2} \ldots$, at each layer $\ell$ (Figure 2) based on recent past inputs. The attention module at layer $\ell$ in our concept discovery module also attends to the stored concept at layer $\ell$. Additionally, the concept storage unit combines similar concepts stored at layer $\ell$ to obtain the concept at layer $\ell + 1$. Using this idea, we propose the following learning method:

1. Given an input vector $x_t$ at time $t$.

2. For each layer $\ell \in [L]$, we store a sequence of $m$ past signatures $T_{\ell,1}, T_{\ell,2}, \cdots, T_{\ell,m}$, where $T_{1,i} = x_{t+1-i}$.

3. We compute the attention (as described later) of $T_\ell(x_t)$ and $T_{\ell,1}, T_{\ell,2}, \cdots, T_{\ell,m-1}$, obtain the $T_{\ell,i_1}, T_{\ell,i_2}, \cdots, T_{\ell,i_K}$ with the highest $K$ attention score, then compute the signature of them to get $T_{\ell+1}(x_t)$. We then update $T_{\ell+1,i} = T_{\ell+1,i-1}$ for $i \geq 2$ and $T_{\ell+1,1} = T_{\ell+1}(x_t)$ to maintain $m$ signatures per layer.

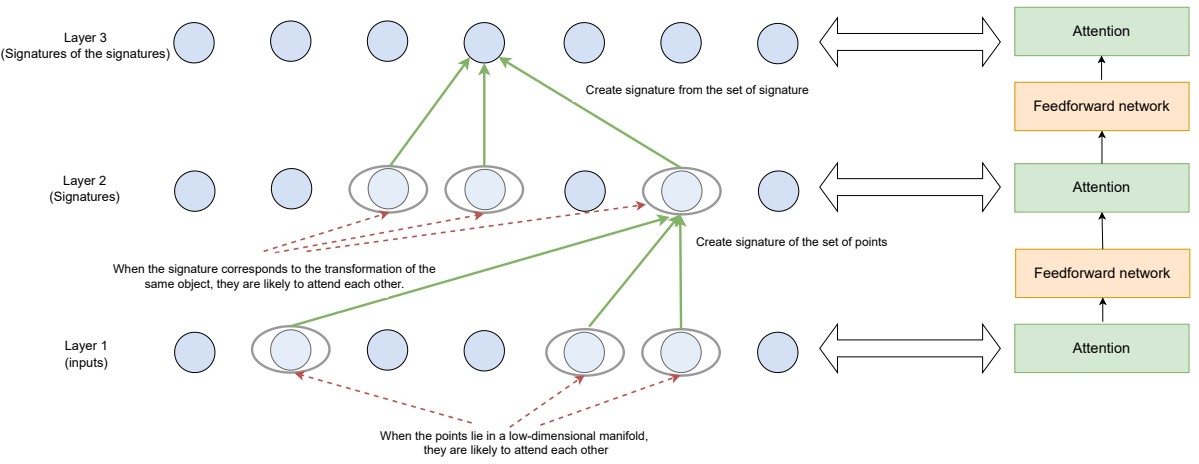

Figure 2: Overview of our learning architecture.

**The mechanism behind the learning method.** Here, the attention is simply based on $\frac{\langle x,y\rangle}{\|x\|_2\|y\|_2}$. We will discuss the mechanism of attention in the next section, at a high level, our results show that:

1. Points lying in a low dimensional manifold are more likely to attend to each other ( Proposition 3.1), so $T_{\ell+1}(x_t)$ will more likely be a signature for the low dimension manifold that is spanned by a subset of $T_{\ell,i_1}, T_{\ell,i_2}, \cdots, T_{\ell,i_K}$. Thus, if there is a subset of signatures in $\{T_{\ell,2}, T_{\ell,3}, ..., T_{\ell,m}\}$ that lies on a low dimension manifold that contains $T_\ell(x_t)$, then $T_{\ell+1}(x_t)$ is more likely to be the signature of that manifold.

2. Signatures corresponding to the transformation of the same object are more likely to lie in a low-dimensional manifold.

As an example, if there is a set of $K-1$ points $x_{i_1}, x_{i_2}, \cdots, x_{i_{K-1}}$ and $x_t$ that lies in the same manifold, then $T_2(x_t)$ is more likely to be the signature of that manifold.

## 3.1 STRUCTURE ACROSS CONCEPTS IS PRESENT IN THE SIGNATURES

We first show how attention based on distance between points can be used to group points from the same concept. We also show for linear manifolds (specifically subspaces) that similar concepts have similar signatures and the signature of intersection of multiple linear manifolds can be obtained from the individual signatures of the manifold.

**Proposition 3.1.** [Attention based on cosine similarity] Given a $k$ dimensional manifold $X$ with constant distortion, for every $\varepsilon > 0$, $N \in \mathbb{Z}^+$, for every set of $\Omega(N(\log(k)/\varepsilon)^k)$ points in $\{x \mid \|x\|_2 \leq 1\}$, there must be a set $S$ of $N$ points such that for every $x, x' \in S$, $\frac{\langle x,x'\rangle}{\|x\|_2\|x'\|_2} \geq 1 - \varepsilon$.

Note that different attention heads could apply attention at different scales of granularity $\varepsilon$.

*Proof.* First, we can consider a $k$-dimensional subspace, then the unit ball in the $k$-dimensional subspace can be covered by $O((\log(k)/\varepsilon)^k)$ rays of angles at most $\varepsilon$. By pigeon-hole principle, we complete the proof. For a general $k$-dimensional manifold with constant distortion, we can map it back to $k$-dimensional subspace which preserves distance up to a constant multiplicative factor. $\square$

The proposition implies that random points on low-dimensional manifolds are much closer to each other in general, compared to random points over the entire space. Thus, as we will show later, points on a single low-dimension manifold are more likely to attend to each other in an attention layer.

Next we show how two the intersection of two concepts can be learned just from the signatures of the undelying concepts. We also show that for two linear manifolds that are subspaces, their signatures have a high dot product if the subspaces have a higher overlap (i.e., the common subspace is of higher dimension).

**Proposition 3.2.** Given two concepts with manifolds $U_1$ and $U_2$, the intersection of the two concepts is given by the intersection of the manifolds $U_1 \cap U_2$. The signature of this intersection can be easily computed as follows. Define $F(U) = I - T(U)$. Then,

$$F(U_1 \cap U_2) = (F(U_1).F(U_2))^\infty$$

Here by the $\infty$ in the power can be thought of as taking a very high power.

Further if they are linear manifolds that are subspaces of dimension $k$ then the similarity between their signatures increases with the dimensionality $\dim(U_1 \cap U_2)$ of their intersection.

1.
$$F(U_1).F(U_2) \geq \dim(U_1 \cap U_2)$$

   Note that $T(U_1).T(U_2) = F(U_1).F(U_2) + d - \dim(U_1) - \dim(U_2)$.

2. In contrast two random subspaces $U_1$ and $U_2$ of dimension $k$ in $d$-dimensional space have a small dot product if $k^2 << d$.
$$E_{U_1,U_2}[F(U_1).F(U_2)] = k^2/d$$

**Lemma 3.1** (Similar Manifolds). Consider a manifold specified by a degree $l$ equation written in the form $c.\Phi(x) = 0$ where $c$ is a (normalized) coefficient vector Given two manifolds $U_1, U_2$ specified by two such coefficient vectors $c_1, c_2$ $T(U_1).T(U_2) = (c_1.c_2)^2$

Next we show that if concepts have a dictionary structure formed from atomic concepts then such a structure can be discovered easily

**Lemma 3.2.** [A Dictionary of Concepts] If the concepts have a structural relationship of the form that there is a set of latent atomic concepts and all concepts are obtained by taking sparse unions of these atomic subspaces, then this structure can easily be discovered from the concept signatures.

*Proof.* This follows from the fact that the signature of the intersection of two concepts can be obtained from the signatures of the individual concepts Proposition 3.2. By repeated taking intersections we get the set of atomic concepts. □

## 3.2 CONCEPTS THAT ARE UNION OF A FEW SIMPLER CONCEPTS

**Remark 3.1.** A stick figure human would consists of about 6 curves drawn on a 2d plane. At the first level we would get signatures for each of those curves. at the second level we would get a signature of the curve signatures. This would become the signature of the entire stick figure. Even though the total degrees of the freedom here is large, one can easily identify the common invariant part of the human stick figure concept from just a few examples.

The following Lemma demonstrates how a higher level concept obtained by combining a few simpler concepts can be represented by the signature of the signatures of the underlying concepts; for example, a rectangle is obtained by a union of four line segments and the human stick figure would consist of six curves.

**Lemma 3.3** (Moment statistics memorize a small set of points). For $k$ points, the signature obtained by the $O(k)$th moment statistic uniquely identifies exactly the set of $k$ points.

*Proof.* This follows from the fact that there is a polynomial of degree $2k$ whose zero's coincide with the set of given $k$ points. This polynomial must be in the null space of the moments statistic matrix. □

# 4  SIGNATURE OF SIGNATURES

As outlined in the introduction, the right notion of signatures and concepts should allow us to build higher-level concepts from lower-level concepts and allow us to identify common traits in objects by even simply intersecting concepts. In this section, we show that our idea of building signatures from statistics can be used to learn simple higher-level concepts from lower-level concepts. We defer the proofs of technical results of the section to Appendix B.

We start with learning the concept of a given circle by looking at the null-space formed by the points on the circle. Next, by looking at the signatures of several circles and the null-space of the moments of these signatures we learn the general concept of a circle. Similarly, we show that rotated versions of an image lie on a manifold which can be used to obtain a rotation invariant representation of images.

Further, the idea of intersecting concepts can be used to define a common direction of motion from several moving objects that share the direction of motion (i.e., we identify the direction of motion as a common concept for these different objects).

## 4.1  SIGNATURE OF A CONCEPT OF A CIRCLE

**Lemma 4.1.** By setting $S(x) = (\phi(x))^{\cdot 2}$, where $\phi(x) = [1; x]^{\cdot 2}$, we obtain an invariant signature of a unit circle $T(X) = I - w w^\top$ where $w = [-1, 0, 0, 1, 0, 1]$. This signature can be obtained even from random points from an arc of the circle.

**Lemma 4.2.** Individual signatures $T(X)$ of several concentric circles lie in a 1-d manifold of degree 2; the signature of these signature corresponds to the the concept of a circle (centered at a origin).

## 4.2  ROTATION INVARIANT SIGNATURE

In this section, we show how rotating an image produces the signatures on an analytic manifold which means we can use Theorem 5.2 to obtain a signature of this manifold.

**Lemma 4.3.** Let $X$ denote the point cloud of the pixels of image (each pixel can be viewed as concatenation of position and color coordinates) and let $X_\theta$ denote the rotation of the image under rotation $\theta$. Then, $S(X_\theta)$ can be written as a Taylor series in $\theta$ and therefore, we can obtain the rotation invariant signature of $X$ from the signature of signatures $\{S(X_\theta)\}_\theta$.

See Appendix B.2 for the Taylor series approximation of the rotation and an additional example of translation invariant signature of $X$.

## 4.3  SIGNATURE OF A OBJECT MOVING IN A SPECIFIC MOTION PATTERN

**Lemma 4.4.** The signatures of the motion of two objects moving with the same velocity function is sufficient to obtain the signature of that velocity function. Any other object with same velocity function will match that signature. Given a set of points $X_1, X_2$ each moving along separate velocity functions $v_1(t), v_2(t)$, will result in a collection of concept signatures for each point's trajectory. All the concept signature of point trajectories from one set will intersect in the corresponding velocity concept signature.

# 5  PROPERTIES OF MANIFOLDS

In this section, we study properties of manifolds and show auxiliary results for polynomials. In particular, we consider the manifold that is implicitly represented by a generator of $k-$dimensional space and show that there exists a polynomial $H(x)$ that represents the manifold. We first start with the case when the generator of the data is low degree polynomial, i.e., $X = G(z)$ where Generator $G : \mathbb{R}^k \to \mathbb{R}^d$ is a degree $r$ polynomial. We also show that polynomials with non-trivial norm are unlikely to be very close to zero; this is useful for checking membership on manifolds. Due to space constraints, we only provide theorem statements in this section and provide proofs in Appendix C.

## 5.1 EXPLICIT POLYNOMIAL REPRESENTATION FROM IMPLICIT POLYNOMIALS

**Theorem 5.1** (Explicit polynomials from implicit polynomials). Suppose $X \in \mathbb{R}^d$ lies in a $k-$dimensional manifold and is represented as a degree $r$ implicit polynomial by $X = G(z)$ where $z \in \mathbb{R}^k$. Then, the manifold can be written as zero sets of $d - k$ polynomials $H(X)$ and the degree of each of the polynomials is at most $r^k$.

The proof uses the fact that when the data generator equation $X = G(z)$ is raised to the sufficiently high tensor power $u$ to obtain $X^{\cdot u} = G(z)^{\cdot u}$, then there will be more different monomials in $X^{\cdot u}$ than $G(z)^{\cdot u}$ because $X$ has more number of variables than $z$. Therefore, one can eliminate the implicit variables $z$ to obtain a polynomial representation corresponding to the manifold.

## 5.2 APPROXIMATE POLYNOMIAL REPRESENTATION FOR ANALYTIC MANIFOLDS

Next, we show that even if $G(z)$ is not a polynomial but a general analytic function, then the manifold can be approximated as $d - k$ polynomials of high-enough degree.

**Theorem 5.2.** Suppose a $k-$dimensional manifold of the data $X \in [-1, 1]^d$ is given by $X = G(z)$ where $G : [-1, 1]^k \to [-1, 1]^d$ is an analytic function and $\|\nabla^{(m)} G(z)\|_2 \leq 1$ for any $m$. Then, there exists a polynomial in $X$, $H(X)$ of degree $r$ such that for points on the manifold, $\|H(X)\|_\infty \leq r^{-\frac{r^{(1+1/k)}}{2c^k k}}$ for some absolute constant $c$.

## 6 CONNECTION TO TRANSFORMER ARCHITECTURE

Our learning architecture includes attention mechanism that groups from the same concept together and MLP layer which computes the signature of the concept and a memory table which keeps track of moment signature seen so far. Even though we presented the signatures as matrices, we will argue how this signatures will arise from flattened vectors from random MLP layers with small amount of trained projection.

On a high level, we envision that a feedforward network can contain the signature $S(x) = \phi(x)\phi(x)^\top$ for each token and attention can be used to combine tokens from the same concept together to obtain $M(X) = \frac{1}{|X|} \sum_{x \in X} S(x)$ and the subsequent feedforward network can be used to obtain $T(X)$ using $M(X)$ because $T(X)$ is the null space of $M(X)$ and it can be obtained by taking by repeated multiplication of $(I - M(X))$. See Figure 3 for the high-level overview of the connection and see Appendix D for the detailed discussion.

In particular, we show that a two-layer neural network with random weights can obtain $M(X)^2$ upto rotation, i.e., there exists a fixed projection of representation produced by the MLP layers that equals $M(X)^2$. One can use this two-layer neural network as a building block to obtain the higher powers of $M(X)$ and the null-space of a high enough powers of $M(X)$ defines the null-space signature of the concept.

We denote the data matrix $\mathbf{X} \in \mathbb{R}^{N \times d}$ where $i^{th}$ row is given by $x_i$. We consider the square activation and the following two-layer neural network with $m$ hidden units whose weights are initialized using standard Gaussian $\mathcal{N}(0, I)$:

$$G_{1,j}(x_i) = \sigma(x_i \cdot r_j), \quad \hat{G}_{1,j} = \frac{1}{N} \sum_{i=1}^{N} G_{1,j}(x_i) \text{ and } G_{2,j} = \sigma(\hat{G}_{1,j})$$

where $\sigma$ is an element-wise square activation. For wide enough MLP layers, we show that $\hat{G}_{1,j}$ is equivalent to $M(X)$ by a projection and $G_{2,j}$ is equivalent to $M(X)^2$ upto a projection.

**Theorem 6.1.** There exists a projection of $\hat{G}_{1,j}$ that gives $M(X)$:

$$E_{r \sim \mathcal{N}(0,I)}[r_j^{(.2)} \hat{G}_{1,j}] - (d + 1)I = M(X).$$

Similarly, there exists a projection of the representation generated by first and second layer such that

$$E_{r \sim \mathcal{N}(0,I)}[r_j^{\otimes 2} \hat{G}_{1,j} + \beta_1 r_j^{\otimes 2} G_{2,j}] + \beta_2 I = M(X)^2$$

where $\beta_1$ and $\beta_2$ only depend on $d$.

The full proof can be found in Appendix D.

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

## A    PROOFS FOR SECTION 3

### A.1    PROOF FOR LEMMA 3.1

*Proof.* If a manifold $U$ is specified by a polynomial equation of degree $l$ and we use $lth$ moments, there will be exactly one eigenvector in the null space specified by $c$. Thus $T(U) = cc'$. So $T(U_1).T(U_2) = c_1 c_1' c_2 c_2' = (c_1.c_2)^2$

$\square$

**Corollary A.1.** [Similar Manifolds]

Consider random hyperplanes and spheres in $d$ dimensions where all coefficient's of hyperplanes are chosen as normal random variables and for spheres the coordinates and radius chosen as unit normal random variables. By looking at the similarity $T(U_1).T(U_2)$ we get

1. The expected similarity between two random lines is $1/d$

2. Between two parallel lines is $1 - O(1/d)$

3. Between two random spheres is $1/5$

4. Between two concentric spheres is $1$

5. Between a random line and a random sphere is $O(1/d)$

### A.2    PROOF OF PROPOSITION 3.2

*Proof.* For the signature of $U_1 \cap U_2$, first consider the case when they linear manifolds that are subpaces. Then $F(U) = I - T(U)$ is the projection matrix for the subspace. $T(U_1 \cap U_2)$ will be the union of the null spaces $T(U_1), T(U_2)$ – this is because the intersection of the concepts will satisfy any polynomial equation satisfied by either of the two concepts. To obtain the projection matrix $F(U_1 \cap U_2)$ we note that repeated multiplication of a vector by $F(U_1)$ and $F(U_2)$ will nullify any component vector outside intersection of $U_1$ and $U_2$; if a vector is in the intersection

then the multiplications keeps it unchanged; otherwise its length keeps decreasing till it ends up getting projected into $U_1 \cap U_2$. The same argument when applied on subspaces formed over $\phi(x)$ which extends this to general manifolds.

Next, let $v_1, .., v_{k_1}$ be a basis for $U_1$ subspace and $w_1, .., w_{k_2}$ be a basis for $U_2$ subspace. Then $F(U_1) = \sum v_i v_i^\top$ and $F(U_2) = \sum w_i w_i^\top$. Thus, $F(U_1).F(U_2) = \sum_{i,j}(v_i.w_j)^2$. If $U_1$ and $U_2$ intersect in $\dim(U_1 \cap U_2)$, then we can find basis that share $\dim(U_1 \cap U_2)$ basis vectors. This gives

$$F(U_1).F(U_2) \geq dim(U_1 \cap U_2).$$

For random $v_i$ and $w_j$, $E[v_i.w_j] = \frac{1}{d}$ which proves the result for random subspaces $U_1$ and $U_2$ (i.e., $E[F(U_1).F(U_2)] = k^2/d$).

The result for $T(U)$ follow from the fact that $T(U) = I - F(U)$.

$\square$

# B  APPENDIX FOR SECTION 4

## B.1  OMITTED PROOFS OF SECTION 4

In this section, we provide the omitted proofs of Section 4.

### B.1.1  PROOF OF LEMMA 4.1

*Proof.* Note that $\phi(x)$ includes all monomials of degree 2 (that is $\phi(x) = [1, x_1, x_2, x_1^2, x_1 x_2, x_2^2]$). The circle signature is given by $T(X) = I - ww^\top$ where $w = [-1, 0, 0, 1, 0, 1]$. This corresponds to the equation of the circle $T(X).S(x) = 0$ or $\phi(x).w = 0$ or $x_1^2 + x_2^2 - 1 = 0$. This is because points from the arc can only satisfy this one equation of degree 2. Instead of $\phi(x) = [1; x]^{\cdot 2}$, if we had used a higher power $l$, then too any polynomial equation satisfied by an arc must be satisfied by the circle and must have $x_1^2 + x_2^2 - 1$ as a factor. $\square$

### B.1.2  PROOF OF LEMMA 4.2

*Proof.* The nullspace of a circle of radius $r$ is given by the equation $x_1^2 + x_2^2 - r^2 = 0$ which corresponds to the singular vector $w = [-r^2, 0, 0, 1, 0, 1]$. Thus, the signature $C_r$ of a circle of radius $r$ is $I - ww^\top$ where $w = [-r^2, 0, 0, 1, 0, 1]$

Thus $C_r = a_1 + a_2 r + a_3 r^2$ where $a_1, a_2, a_3$ are constant vectors when flattened. Thus we can think of $C_r$ as isomorphic to $[z_0, z_1, z_2] = [1, r^2, r^4]$. Thus under this isomorphic view $C_r$ is the intersection of the polynomial equations: $z_0 - 1 = 0, z_2 - z_1^2 = 0$

Thus $T(z) = T(C_r^{\cdot 2}) = z^{\cdot 2}(I - w_1 w_1' - w_2 w_2')$ where $w_1.z = 0$ and $w_2.z = 0$ capture the above two equations. This is because if we take $[1, r^2, r^4]$ for a few different values of $r$ any degree 4 equation $z^{\cdot 2}$ satisfies is satisfied by all such $[1, r^2, r^4]$.

$\square$

### B.1.3  PROOF OF LEMMA 4.4

*Proof.* Different objects moving with same velocity can be viewed as subspaces that intersect in the common velocity subspace. For e.g. an object moving in 3d space can be viewed as 1d manifold. And if the velocity is common these are parallel manifolds. By appending a 1 coordinate these can be seen as 2d manifolds that intersect in the common velocity manifold. By Proposition 3.2 two such signatures are sufficient to obtain the signature $F(V)$ of the velocity manifold. Checking if another object motion $F(O)$ has the same velocity is also easy: check if $F(O)F(V) == F(V)$.

$\square$

## B.2  SIGNATURE OF TRANSLATED AND ROTATED IMAGE MANIFOLD

In this section, we show how rotating an image produces the signatures on an analytic manifold which means we can use Theorem 5.2 to obtain a signature of this manifold. Here, we explicitly show the Taylor series that transforms signatures under rotation $\theta$.

**Lemma B.1.** Let $X$ denote the point cloud of the pixels of image (each pixel can be viewed as concatenation of position and color coordinates) and let $X_\theta$ denote the rotation of the image under rotation $\theta$. Then, $S(X_\theta)$ can be written as a Taylor series in $\theta$. (We derive this Taylor series upto the second order approximation).

*Proof.* Note that each point goes through following transformation:

$$x' = x \cdot \cos\theta + y \cdot \sin\theta \quad \text{and} \quad y' = -x \cdot \sin\theta + y \cdot \cos\theta,$$

where point $x'$ is obtained by applying the rotation matrix corresponding to $\theta$ angle on point $x$. The second degree moment can be computed as

$$M_{x'} = \mathbb{E}[x'] = M_{x'} \cos\theta + M_{y'} \sin\theta$$
$$M_{y'} = \mathbb{E}[y'] = -M_x \sin\theta + M_y \cos\theta$$
$$M_{x'^2} = \mathbb{E}[x'^2] = M_{x^2} \cos^2\theta + M_{y^2} \sin^2\theta + 2M_x M_y \cos\theta \sin\theta.$$
$$M_{y'^2} = \mathbb{E}[y'^2] = M_{x^2} \sin^2\theta + M_{y^2} \cos^2\theta - 2M_x M_y \cos\theta \sin\theta$$
$$M_{x'y'} = \mathbb{E}[x'y'] = -M_{x^2} \sin\theta \cos\theta + M_{y^2} \sin\theta \cos\theta + M_x M_y (\cos^2\theta - \sin^2\theta)$$

For small rotation $\theta$, we can approximate $\sin\theta \approx \theta$ and $\cos\theta \approx 1 - \frac{\theta^2}{2}$ using taylor series with approximation error of $O(\theta^3)$. Using this approximation, we obtain:

$$M_{x'} = M_x\left(1 - \frac{\theta^2}{2}\right) + M_y\theta$$

$$M_{y'} = -M_x\theta + M_y\left(1 - \frac{\theta^2}{2}\right)$$

$$M_{x'^2} = M_{x^2}(1 - \theta^2) + M_{y^2}\theta^2 + 2M_{xy}\theta$$

$$M_{y'^2} = M_{x^2}\theta^2 + M_{y^2}(1 - \theta^2) - 2M_{xy}\theta$$

$$M_{x'y'} = -M_{x^2}\theta + M_{y^2}\theta + M_{xy}(1 - 2\theta^2).$$

This second order moment allows us to use the power transformation to get the signature of the rotated data manifold. Simplifying above equations, we obtain

$$\begin{pmatrix} M_{x'} \\ M_{y'} \\ M_{x'^2} \\ M_{y'^2} \\ M_{x'y'} \end{pmatrix} \approx \begin{pmatrix} M_x \\ M_y \\ M_{x^2} \\ M_{y^2} \\ M_{xy} \end{pmatrix} + \theta \begin{pmatrix} M_y \\ -M_x \\ M_{xy} \\ -M_{xy} \\ M_{y^2} - M_{x^2} \end{pmatrix} + \frac{\theta^2}{2} \begin{pmatrix} -M_x \\ M_y \\ 2M_{y^2} - 2M_{x^2} \\ 2M_{x^2} - 2M_{y^2} \\ -4M_{xy} \end{pmatrix}$$

$\square$

Next, we describe the manifold obtained by translation of a given image.

**Lemma B.2** (Signature of translated image manifold)**.** Let $S(X_{(u,v)})$ be the signature of an image shifted by $(u, v)$. Then, all such signatures $S(X_{(u,v)})$ over different shifts lies on an analytic manifold.

*Proof.* Suppose each point goes through the following transformation:

$$x' = x + u \quad \text{and} \quad y' = y + v$$

The moments after the translation is given by

$$M_{x'} = M_x + u$$
$$M_{y'} = M_y + v$$
$$M_{x'^2} = M_{x^2} + 2M_x u + u^2$$
$$M_{y'^2} = M_{y^2} + 2M_y v + v^2$$
$$M_{x'y'} = M_{xy} + M_x v + M_y u + u \cdot v$$

Suppose the translation $u$ and $v$ are small such that the second order terms are neglible (e.g., $u^2$, $v^2$ and $u \cdot v$ are negligible), then we can obtain the second order moment statistics as a linear approximation with quadratic approximation error as follows:

$$
\begin{pmatrix} M_{x'} \\ M_{y'} \\ M_{x'^2} \\ M_{y'^2} \\ M_{x'y'} \end{pmatrix} \approx \begin{pmatrix} M_x \\ M_y \\ M_{x^2} \\ M_{y^2} \\ M_{xy} \end{pmatrix} + u \begin{pmatrix} 1 \\ 0 \\ 2 \cdot M_x \\ 0 \\ M_y \end{pmatrix} + v \begin{pmatrix} 0 \\ 1 \\ 0 \\ 2 \cdot M_y \\ M_x \end{pmatrix}
$$

$\square$

## C  PROOFS FROM SECTION 5

### C.1  EXPLICIT POLYNOMIAL REPRESENTATION FROM IMPLICIT POLYNOMIALS

We will first prove the case when $X$ is a $k-$dimensional manifold in $(k+1)-$dimensional space.

**Lemma C.1.** Suppose $X \in \mathbb{R}^{k+1}$ can be represented as a degree $l$ polynomial in terms of implicit random variables $z \in \mathbb{R}^k$. i.e., $X = G(z)$ where the degree of $p$ is $r$, then there exists a polynomial in $X$, $H(X)$, of the degree $(cr)^k$ for some absolute constant $c$ such that $H(X) = 0$ for points on the manifold.

*Proof of Lemma C.1.* Let $u$ be the upper bound on the degree of $H(X)$. We show that there exists a polynomial constructed from the entries of $X^{\otimes u}$ such that $H(X) = 0$ for points on the manifold.

We have $X^{\otimes u} = G(z)^{\otimes u}$ by taking $u^{\text{th}}$ tensor power of $X = G(z)$ equality. Denote the number of different monomial entries in $X^{\otimes u}$ with $X \in \mathbb{R}^{k+1}$ by $m(u, k+1)$. Observe that $X^{\otimes u} = G(z)^{\otimes u}$ has $m(u, k+1)$ different equalities from because $X^{\otimes u}$ has $m(u, k+1)$ different monomials. The right side $G(z)^{\otimes u}$ has at most $m(u \cdot r, k)$ different monomial terms, and when $m(u, k+1) > m(u \cdot r, k)$, we can eliminate the implicit variables in the remaining $m(u, k+1) - m(u \cdot r, k)$ equations. The remaining equations are consistent because $X$ is generated using $G(z)$.

The degree of the polynomial $\phi(X)$ is the smallest $u$ such that $m(u, k+1) > m(u \cdot r, k)$. The number of monomials of degree $u$ in $k+1$ variables is given by $m(u, k+1) = C_u^{u+k}$. The degree of $\phi(X)$ is smallest $u$ such that $C_k^{u+k} > C_{k-1}^{r \cdot u+k-1}$. This inequality satisfies when $u > (cr)^{k-1}$ for some absolute constant $c$.

$$
C_{k-1}^{r \cdot u+k-1} \leq \frac{e^{k-1}(r \cdot u + k - 1)^{k-1}}{(k-1)^{k-1}} \leq \frac{(2e)^{k-1}(r \cdot u)^{k-1}}{(k-1)^{k-1}} \leq \left( \frac{2e}{c(k-1)} \right)^{k-1} u^k
$$

$$
\leq \left( \frac{1}{k^k} \right)(u+k)^k \leq C_k^{u+k}
$$

The second inequality follows from $k - 1 \leq r \cdot u$. $\square$

*Proof of Theorem 5.1.* For extending the Lemma C.1 to $X \in \mathbb{R}^d$ case, we start by identifying $k$ direction along a local neighbourhood of any point on the neighbourhood. Note that this can be done by looking at the tangent direction of the manifold of the points in the neighbourhood. After determining the tangent direction, we can rotate the points so that $\{X_1, X_2, \ldots, X_k\}$ in the neighbourhood.

Assuming $\{X_1, X_2, \ldots, X_k\}$ covers a $k-$dimensional space, we use Lemma C.1 for each of $\{X_{k+1}, \ldots, X_d\}$ as $k + 1$-th coordinate and obtain $d - k$ polynomials of the degree $(cl)^k$ for some absolute constant $c$. Note that each of the polynomials contains a feature mapping $H(X)$ that depends on the first $k$ coordinates of $X$ and one of the last $d - k$ coordinates such that $H(X) = 0$ for points on the manifold. $\square$

### C.2  APPROXIMATE POLYNOMIAL REPRESENTATION FOR ANALYTIC MANIFOLDS

We will first prove for the case when $X$ is a $k-$dimensional manifold in a $(k+1)-$dimensional space given by the analytic function.

**Lemma C.2.** Suppose the data $X \in [-1, 1]^{k+1}$ lies in a $k-$dimensional manifold by equation $X = G(z)$ where $G : [-1, 1]^k \to [1, 1]^{k+1}$ is an analytic function with bounded gradient $\|\nabla^{(m)} G(x)\| \leq 1$. Then, there exists a polynomial in $X$, $H(X)$ of degree $u$ such that for points on the manifold, $|H(X)| \leq u^{-\frac{u^{(1+1/k)}}{2c^k k}}$.

*Proof of Lemma C.2.* Since $G(z)$ is an analytic function with the bounded derivatives so we can approximate $G(z)$ using a degree $q$ Taylor approximation $G'(z)$ within the ball of a unit radius of $z$ with the approximation error $\frac{1}{q!}$. Now, instead of Taylor expanding each term in $G(z)$, we can expand each term in $G(z)^{\otimes l}$. Suppose for any multi-index $\alpha = (\alpha_1, \dots, \alpha_l)$, the Taylor approximation of $G_\alpha(z) = \prod_{i=1}^l G_{\alpha_i}(z)$ of degree $q$ is given by $\tilde{G}_\alpha(z)$ as follows:

$$\tilde{G}_\alpha(z) = G_\alpha(\mathbf{0}) + \sum_{i=1}^{q-1} \frac{\langle \nabla^{(i)} G_\alpha(\mathbf{0}), z^{\cdot i}\rangle}{i!}.$$

where $\mathbf{0}$ represents the zero vector. Using the remainder theorem of the Taylor series, we have

$$|\tilde{G}_\alpha(z) - G_\alpha(z)| \leq \frac{\|\nabla^{(i)} G_\alpha(\mathbf{0})\|}{q!} \leq \frac{l^q}{q!}.$$

where the last inequality follows from the fact that the $i^{\text{th}}$ derivative of $\tilde{G}_\alpha(z)$ for any $\alpha$ can be written as a sum of $l^i$ terms with each term bounded by 1. Hence, we can approximate each term in $G(z)^{\otimes l}$ with $\tilde{G}$ within the ball of the unit radius with the approximation error $l^q/q!$ where each element of $\tilde{G}(z)^{(\cdot u)}$ is a degree $q-$polynomial obtained by Taylor expansion.

Recall that the $m(l, k)$ is defined as the number of different monomials of degree $l$ in $k$ variables. The total number of variables in $\tilde{G}(z)$ is given by $m(q, k)$ and the total number of different equations in $X^{\cdot u} = G(z)^{\cdot u}$ is $m(u, k + 1)$. The degree of polynomial $H(X)$ is the smallest $l$ such that $m(u, k + 1) > m(q, k)$. This inequality reduces to $C_k^{u+k} > C_{k-1}^{q+k-1}$. Choosing $u = c^{k-1} q^{\frac{k-1}{k}}$ for some absolute constant $c$, then this inequality holds.

$$C_{k-1}^{q+k-1} \leq \left(\frac{e(q+k-1)}{(k-1)}\right)^{k-1} \leq \frac{u^k}{k^k} \leq \frac{(u+k)^k}{k^k} \leq C_k^{u+k}$$

Next, we show that the polynomial is close to zero for the points on the manifold. The polynomial obtained by a Taylor expansion of the analytic function $G$ with the truncation at $q^{th}$ degree has approximation $\frac{u^q}{q!}$. Putting the value of $q = \frac{u^{k/(k-1)}}{c^k}$, we obtain the approximation error to be $\frac{u^q}{q!} \leq \frac{c^q u^q}{q^q} = c^q u^{-\frac{q}{k}} = u^{-\frac{u^{(1+1/k)}}{2c^k k}}$. $\qquad\square$

*Proof of Theorem 5.2.* Similar to proof of Theorem 5.1, we start by rotating the coordinates at each point such that the first $k$ coordinates $\{X_1, X_2, \dots, X_k\}$ in the neighbourhood covers $k$ dimensional space. Then, we use Lemma C.2 for each of $\{X_{k+1}, \dots, X_d\}$ as $k + 1$-th coordinate and obtain $d - k$ polynomials that satisfies the given property. $\qquad\square$

## C.3  QUANTITATIVE BOUNDS FOR CHECKING MEMBERSHIP

We will use the following classical result about multi-variate polynomials:

**Lemma C.3** (Anti-concentration for polynomials, Carbery & Wright (2001)). *Let $P : \mathbb{R}^n \to \mathbb{R}$ be a degree at most $d$ polynomial. Let $x$ be a random point in the unit ball. Let $\|P\|_2 = E_x[P(x)^2]^{1/2}$. Then, for any $t, \epsilon$,*

$$Pr_x[|P(x) - t| < \epsilon\|P\|_2] = O(\epsilon^{1/d}).$$

The following lemma says if we start from a polynomial with non-trivial norm, then for any ball that is centered not too far from the origin, the polynomial is unlikely to be very small. It will be important that this probability goes to zero as the radius of the ball increases.

**Lemma C.4.** *Let $Q : \mathbb{R}^k \to \mathbb{R}$ be a degree $s$ polynomial such that the sum of squares of the coefficients is 1. Then, for any $y \in \mathbb{R}^k$, if $\|y\| \leq R$, and $z$ a uniformly random point in the unit ball of $\mathbb{R}^k$,*

$$Pr[|Q(y + \Delta z)| > \epsilon] < (ks)\epsilon^{1/s}\alpha^{1/s},$$

*where $\alpha = \max((1 + \|y\|_\infty)/\Delta, ((1 + \|y\|_\infty)/\Delta)^s)$.*

*Proof Sketch for Lemma C.4.* Let $P(z) = Q(y + \Delta z)$. We first argue that $E_z[P(z)^2]$ is non-trivially large. First, observe that $E[Q(z)^2] = s^{-O(s)}$ as $Q$'s sum-of-squares of coefficients is 1.

Without loss of generality, let us suppose $P$ has no constant term. Now, suppose that $E[P(z)]^2 = \delta$. Let $P(z) = \sum_S c_S h_S(z)$ where $h_S$ are the Legendre polynomials that form an orthonormal basis for polynomials under the uniform distribution on the sphere. Thus, $\sum_S c_S^2 = \delta^2$.

Now, for $z'$ uniformly random on the unit ball, let $Q(z') = P((z' - y)/\Delta)$. Therefore, $E[Q(z')^2] = E[(\sum_S c_S h_S((z' - y)/\Delta))^2]$.

However, note that the Legendre polynomials of degree $s$ are bounded: in particular for any point $x$, and $h_S$ of degree $s$, we have $|h_S(x)| \le s^{O(s)}(\|x\|_\infty + \|x\|_\infty^s)$.

Therefore, if we let $\alpha = \max((1 + \|y\|_\infty)/\Delta, ((1 + \|y\|_\infty)/\Delta)^s)$, we get
$$E[Q(z')^2] \le (\sum_S |c_S|) \cdot s^{O(s)}\alpha \le k^{O(s)}s^{O(s)}\alpha\delta.$$

Therefore, we get $\delta > (ks)^{-O(s)}/\alpha$.

Finally, applying the previous lemma, we get
$$Pr_z[|P(z)| < \epsilon] = O((\epsilon/\delta)^{1/s}).$$

Therefore, $Pr_z[|Q(y + \Delta z)| < \epsilon] < (ks) \cdot \alpha^{1/s}\epsilon^{1/s}$.

Finally, note that $\alpha = \max((1 + \|y\|_\infty)/\Delta, ((1 + \|y\|_\infty)/\Delta)^s)$ is a decreasing function in $\Delta$. $\qquad\square$

## D  APPENDIX FOR CONNECTION TO TRANSFORMER ARCHITECTURE

We showed that keeping track of signature matrices for concepts allows an easy way to manipulate the concepts and also obtain signatures of higher-level concepts. In this section, we provide details about the connection of our learning architecture to the transformer model and proof of Theorem 6.1.

A feedforward network can be thought of transforming the input $x$ to obtain the feature transform $\phi(x)$. In Proposition 3.1, we showed that the cosine similarity of points from the same manifold is high. A similar argument also holds for the feature transform $\phi(x)$ therefore cosine similarity-based attention mechanism will group points from the same concept together and therefore will obtain average across points from the same manifold $\hat{G}_{1,j}$. In Theorem 6.1, we show that the $\hat{G}_{1,j}(X)$ contains the flattened version of the kernel signature $M(X)$. Recall that the null-space signature $T(X)$ can be obtained using repeated multiplication of $(I - M(X))$ therefore, in Theorem 6.1, we prove that the subsequent feedforward network contains $M(X)^2$ matrix. Combining the output of attention and feedforward network, we can obtain $(I - M(X))^2$. Repeating such operations will help in obtaining higher powers of $(I - M(X))$ and hence, the null-space signature $T(X)$.

Next, we provide the proof of Theorem 6.1.

*Proof.* Without loss of generality, we assume that $E[X_i^2] = 1$. By definition, $\hat{G}_{1,j} = \frac{1}{N}\sum_{i=1}^N \sigma(x_i \cdot r_j) = \frac{1}{N}\sum_{i=1}^N (r_j^\top x_i x_i^\top r_j) = r_j^\top M(X) r_j$. The expectation of $\hat{G}_{1,j}$ with respect to random Gaussian weights $r_j$ can be simplified using Stein's lemma. Therefore, we have
$$\mathbb{E}_r[r^{\otimes 2}(r^\top M(X)r)] = \mathbb{E}_r[\nabla^{(2)}(r^\top M(X)r) + (r^\top M(X)r)I] = M(X) + (d+1)I.$$
Similarly, simplifying $r_j^{\otimes 2}\hat{G}_{2,j}$ with respect to random Gaussian weights $r_j \sim \mathcal{N}(0, I)$, we have
$$\begin{aligned}
\mathbb{E}[r^{\otimes 2}\hat{G}_{2,j}] &= \mathbb{E}[\nabla^{(2)}(r^\top M(X)r)^2 + (r^\top M(X)r)^2] \\
&= \mathbb{E}[2(r^\top M(X)r)(M(x) + I) + (M(X) + I)rr^\top(M(X) + I) + (r^\top M(X)r)^2 I]. \\
&= 2d(M(X) + I) + M(X)^2 + 2M(X) + I + 3\sum_{i=1}^d M_{i,i}(X)^2 I + \sum_{i \ne j} M_{i,i}(X)M_{j,j}(X)I \\
&= M(X)^2 + (2d + 2)M(X) + (d^2 + 4d + 1)I
\end{aligned}$$

Setting $\beta_1 = -(2d + 2)$ and $\beta_2 = d^2 + 1$, we obtain the result. $\qquad\square$

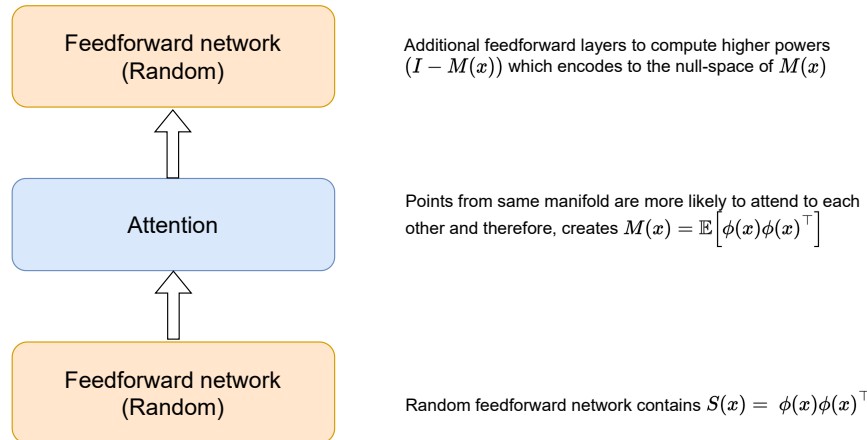

Figure 3: Connection of our learning architecture to the original transformers. A combination of feedforward network and attention module will generate $M(X)$ and a subsequent feedforward layer can generate the nullspace signature $T(X)$ by taking higher power of $I - M(X)$. See Theorem 6.1 for more details.

## E  EXPERIMENTS

In this section, we show the effectiveness of keeping the concept storage unit to represent and manipulate the concepts at different layers using synthetic data.

**Experimental setup.**  We experiment using two synthetic datasets – 1) $n$-circles and 2) $n$-circles+$n$-parabolas. The $n$-circles dataset contains $n$ concentric circles of radius $\{1, 2, \ldots, n\}$. The $n$-parabolas dataset contains $n$ parabolas with the same axis of symmetry but the shift along the axis is $\{0.5, 1.5, \ldots, n - 0.5\}$. The $n$-circles+$n$-parabolas contain $2n$ shapes ($n$ from each of the circle and parabola) and the number of points from each shape is equally divided. In our synthetic dataset, a circle with a given radius (similarly parabola with a given shift) denotes a lower-level concept and all circles with different radius (or parabolas with different shifts) are part of a higher-level concept of the circle shape (or the parabola shape). Note that we chose the shifts of the parabola such that there is an intersection/overlap between the circles and the parabola. See Figure 4a for the example of the 4-circles + 4-parabolas dataset.

As the novelty of our work from the architecture perspective mainly lies in manipulating concepts from the concept discovery module, we focus on experimenting with the concept discovery module with the predefined features $\phi(\cdot)$. For all our experiments, the feature vector contains $\phi(x)$ all monomials up to degree 2. Then, we pass $\phi(x)$ as an input to our learning architecture mentioned in Section 3. We use cosine similarity-based attention on the signatures to combine signatures from the same concept and obtain a higher-level signature.

In experiments of $n$-circles dataset, we use concept storage unit with 2 layers and in experiments of $n$-circles+$n$-parabolas dataset, we use the concept storage unit with 3 layers. In all our experiments, we use $K = 5$. We vary the signature storage size in $\{4 \times 10^3, 4.5 \times 10^3, 5 \times 10^3, 5.5 \times 10^3\}$ and the number of concept $n$ in set $\{4, 6, 8\}$. For all the experiments, we used 100 test samples to calculate all the metrics. All the experimental results are averaged over 5 independent iterations.

**Questions and evaluation.**  We perform experiments to answer the following three questions:

1. To obtain a higher-level concept signature, do signatures only attend to the signatures from the same concept? To answer this question, we measure the percentage of signatures from the same concept in the top $K$ signatures for each data point at all layers in the concept discovery unit where $K$ is the number of signatures to attend to and we present experimental results in Table 1.

2. How does the performance change as we increase we increase the number of concepts? To answer this question, we measure the percentage of the same concept signatures during the attention by increasing $n$ in both synthetic data. We present the findings of our experiments in Table 1.

| Dataset | Layer | Percentage (%)
(Signature storage=4000) | Percentage (%)
(signature storage=5000) |
|---|---|---|---|
| 4-circles | | 98.8 | 99.0 |
| 6-circles | Layer 2 | 96.2 | 98.0 |
| 8-circles | | 96.0 | 97.6 |
| 4-circles+4-parabolas | | 97.9 | 98.3 |
| 6-circles+6-parabolas | Layer 2 | 97.2 | 98.9 |
| 8-circles+8-parabolas | | 96.9 | 97.2 |
| 4-circles+4-parabolas | | 99.5 | 99.7 |
| 6-circles+6-parabolas | Layer 3 | 99.4 | 99.3 |
| 8-circles+8-parabolas | | 98.9 | 98.7 |

Table 1: Percentage of signatures from the same concept in the top $K$ signatures with high attention.

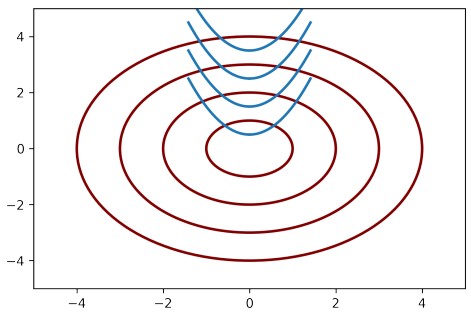

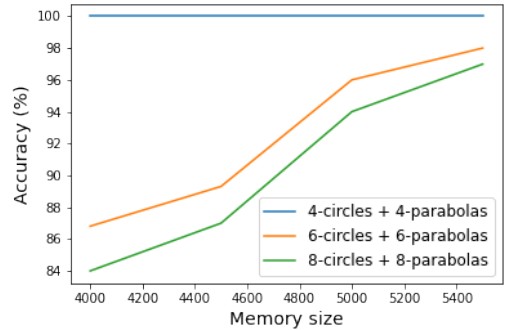

(a) 4-circles + 4-parabolas dataset. Each circle corresponds to a lower-level concept. All four circles are part of a general higher-level concept of circles (similarly for parabola).

(b) We plot the percentage of top-$K$ signatures with the highest attention which corresponds to the signature of a general circle for a new unseen circle signature.

Figure 4: Synthetic dataset of type $n$-circles+$n$-parabola and experimental findings on it.

3. Can a higher-level signature of a concept generalize to unseen concepts from the same lower-level concepts? To be specific, in the $n$-circles + $n$-parabolas dataset, the general signature of a circle is created using signatures of circles with radius $\{1, 2, \ldots, n\}$. Now, we ask the following question: does the signature of a circle with a radius outside of the training radius set have high attention to the general signature of a circle than the general signature of the parabola? To answer this question, we randomly sample a radius from $[1, 10]$ and create a signature of the circle with this radius. Then, we find the top-$K$ signatures with the highest attention in layer 3 and calculate the percentage of them that correspond to the signature of a general circle and report it in Figure 4b.

**Results.** In Table 1, we see that our proposed architecture only attends to the signatures from the same concept class not only in the second layer but also in the third layer. This shows that signatures of circles with different radius only attends to each other and creates a general signature of the circles with different radius (See Figure 2 for details about the architecture). We also see that increasing the number of concepts by increasing $n$ only mildly degrades the performance. Figure 4b shows that having a modular architecture for each concept helps in associating a new unseen lower-level concept to the correct higher-level concept and quickly learning about the new concept.

