# OpenReview forum: "Simple mechanisms for representing, indexing and manipulating concepts"
_ICLR.cc/2024/Conference — Submitted to ICLR 2024_

### Official Review · Reviewer_ViNR · 2023-10-31

**Soundness:** 3 good
**Presentation:** 3 good
**Contribution:** 3 good
**Rating:** 8
**Confidence:** 3

**Summary:**

This paper presents an approach for concept learning, that is based on the idea of generating concept signature from the matrix of moment statistics. Such signatures can be recursively combined in order to hierarchically construct higher-level concepts. The methodology is based on manifold learning via an attention mechanism, and a dictionary of concept signatures can be exploited as a sort of memory of already learned concepts.

**Strengths:**

+ Concept learning is a very important and challenging problem, still far from being solved. Advances in this area are crucial for the development of AI.

+ The paper presents a theoretical methodology grounded on manifold learning. The formalization is sound, as far as I could check.

**Weaknesses:**

- The work is mostly theoretical, with neither experiments nor examples that could drive the reader through the methodology.

- The paper does not describe how the proposed approach could be implemented in practice, i.e., for example illustrating how and whether it could scale up to a large number of concepts and manifolds, or to what extent it would be possible to associate symbols to the learned concepts.

- Section 6 very briefly explains a link between the proposed approach and transformer architectures, due to the use of an attention mechanism within the manifold. This connection with transformers should probably be made more explicit, as it is not easy to grasp from Section 6.

- The paper is well-written, but also not very easy to follow due to the complex formalism.

**Questions:**

- Pag. 5, please check the use of $l$ vs. $\ell$.
- Pag. 6, Figure 2: "Points lies" -> "Points lie"

---

> ### Author Response · Authors · 2023-11-15
> **Additional information about results**
>
> We thank the reviewer for their time and effort in providing feedback on our work.
>
>
> **Comments on examples or experiments**: In section 4, we show how to obtain the signature of a concept using circles as an example. We also show that by combining signatures of multiple lower concepts (i.e., signatures of concentric circles with different radius), one can obtain the signatures of a higher-level concept (i.e., a general concept of concentric circles). We also show how simple operations on top of signatures yield interesting properties (e.g., rotation and translation invariant signatures of the manifold). If the reviewer can point to any particular part of the paper that would improve with an example, we will try our best to provide such an example.
>
> **Details about the proposed architecture**: We have updated Section 3 to include more information about our proposed architecture and also updated Figure 2 to include more details. Our analysis points to an architecture with a “concept discovery unit” (which is similar to a traditional Transformer but likely smaller as is not used to remember concepts) and a separate “concept dictionary” which remembers all the concepts found in the training data. This separation helps to scale up the memory and number of concepts learned efficiently. Further, the nullspace concept signatures could add to the level of interpretability of the concepts present in an input.
>
> **Connection to Transformer**: We have added a figure (Figure 3) to explain the connection of our proposed learning architecture to transformer architectures. On a high level, a random feedforward network can contain the signature $S(x) = \phi(x) \phi(x)^\top$ for each token and attention can be thought of as combining tokens from the same concept together to obtain $M(X)$ and the subsequent feedforward network can be used to obtain $T(X)$ using $M(X)$ because $T(X)$ is the null space of $M(X)$ and it can be obtained by taking by repeated multiplication of $(I-M(X))$. We have added this discussion to the updated paper.
>
>
> We hope that our response resolves the concerns of the reviewer and we are happy to discuss more to increase the confidence of the reviewer in our work.

---

> > ### Comment · Reviewer_ViNR · 2023-11-21
> > **Thank you for the clarifications**
> >
> > Thank you for the clarifications, for better explaining the architectures and the connections with transformers, and for the changes in the paper. I have kept my score, while slightly increasing my confidence, also after reading the other reviews.

---

### Official Review · Reviewer_UZQy · 2023-11-01

**Soundness:** 2 fair
**Presentation:** 1 poor
**Contribution:** 1 poor
**Rating:** 3
**Confidence:** 3

**Summary:**

The authors proposed a new mathematics to represent and learn concepts. Learning a concept is carried out by its moment statistics matrix, which generates a concrete representation or signature of that concept. Authors also show how higher-level concepts can be developed and how relations to Transformer is also presented.

**Strengths:**

Representing or grounding concepts in the form of geometric manifolds is a promising approach to machine learning and artificial intelligence.

**Weaknesses:**

Unfortunately, this work in the current presented version is still too early for a formal publication. Some mathematical formulas are not clear. The major limitation is the missing of experiments. If the relation to Transformer can be established, can the new method outperform Transformers in some tasks?

The writing of "Proposition 1.1. (Informal summary of results)" is strange. By "Proposition" we expect formal results, but followed by "(Informal summary of results)".

**Questions:**

1. in Definition 2.1, is there any restriction between d and m? For example, d > m or d < M.
2. think of the character recognition (MNIST). There are ten concepts: 0, 1, 2, ..., 9.  Given images of characters x_1, x_2, ..., Each image has the size of 28X28 pixels. How should the kernel signature be defined?
3. in Definition 2.3, P is a mapping from R^m to R^{d-k}, but, P(X) = 0 means mapping x in R^m to R^1.  What does \mathtt{S} mean?
4. In the mechanism behind the learning method. The attention is simply based on the cosine similarity. Does this have enough representation power to distinguish manifolds located at the same orientation but different distances to a reference manifold?
5. Have you ever done any experiments with your new methods?

---

> ### Author Response · Authors · 2023-11-14
>
> We thank the reviewer for their time and effort in providing feedback on our work.
>
> **Answers to weaknesses**
>
> As the reviewer points out, our formalism does suggest a modified transform architecture that includes a separate concept storage unit (dictionary of concepts) and a concept discovery unit, which is much smaller than today’s giant transformers. See Figure 2 in the modified pdf. Thus, our theory formalizes the importance of retrieval-augmented transformer architectures.
>
> Proposition 1.1 is to describe the main results before going into the detailed statements presented later in the paper. We can change the name ‘Proposition 1.1’ to “Informal summary of results”.
>
> **Answers to questions:**
> 1. There is no restriction of parameters in the definition, but typically we would have $d < m$ (feature expansion by the kernel).
> 2. We note that the kernel signature is defined specific to a dataset and the choice of the kernel phi. For MNIST, one could view each image as a point cloud where each point represents (pixel-coordinates, pixel-value). You could now compute the kernel signature for each image $M(x) = \mathbb{E}[ \phi(x) \phi(x)^\top ]$ where expectation over a group of related points that attend to each other. As explained in Section 3.1, the attention will cause points from the same continuous curve to attend to each other. If the image had only one curve (e.g., an image of number “0”), then this would result in a signature for that curve. An image containing multiple curves (such as the image of number “7”) would result in two signatures at the second layer. The third layer will group these two curves into one signature (see section 3.2) and this signature can be used to identify similar images.
> 3. The notation $\mathtt{S}^{d-1}$ denotes the unit-sphere in $d$-dimensions, and we will add this clarification.
> 4. The learning method uses a previously computed representation (such as $\phi(x)$) and applies attention on $\phi(x)$ based on cosine similarity (see figure 3 for more details where the attention is actually over $S(x)$ which is $\phi(x) \phi(x)^\top $. Thus, $S(x).S(y) = (\phi(x).\phi(y))^2 \ )$. As the reviewer points out, cosine similarity cannot distinguish manifolds with the same orientation but different distances but since we are using the kernel transform, they can be distinguished by a suitable kernel (for example, if the distance to a reference point is included as one of the features in the kernel).
> 5. We agree with the reviewer that the present paper lacks experimental work. The paper aimed to take a first-principles approach to the challenging problem of formalizing the recognition of concepts and deduce the implications of such a formalism. While there are several empirical investigations of such questions for existing architectures and methods, the formalism of concepts is quite poorly understood, and it would help to develop theoretical formalism first without being bogged down by the noise in data, variations in architectures, training methods, and datasets. Designing extensive experiments and developing the framework for real-world datasets will be a substantial body of work by itself, just as developing the formalism here was.
>
>
>
>
> We hope that our response resolves the concerns of the reviewer and we are happy to discuss more to clarify any further concerns.

---

> > ### Comment · Reviewer_UZQy · 2023-11-19
> >
> > Thanks. The responses remove part of my concerns and also admit the importance of experiments. I agree with the promising direction of this research, but, experiments are necessary and need to be done and reported (with analysis). When you do experiments, you will test the kernel functions that you described in point 4. Actually, this is one of the key parts of your theory. You may even question whether the cosine function is optimal.

---

### Official Review · Reviewer_G7CX · 2023-11-01

**Soundness:** 3 good
**Presentation:** 3 good
**Contribution:** 2 fair
**Rating:** 6
**Confidence:** 3

**Summary:**

*Note that mathematical claims were not checked in detail*

The authors propose a formalism for hierarchical concept discovery, in which concepts lie on low-dimensional polynomial manifolds whose moment statistics can be used to identify the "signatures" of concepts. They show that in such a setting, the inner product between points on the same concept manifold will be lower than those on different manifolds, such that concept membership can, in principle, be performed with an inner product check. Furthermore, they show that random projections can be used to lower the sample complexity of membership checks to be quadratic in the effective number of dimensions of a manifold.

These theoretical results are then applied to a (theoretical) modification of the transformer architecture, in which the polynomial manifolds correspond to distinct conceptual subspaces (in much the same way the Mechanistic Interpretability community have conceptualized the transformers in the "residual stream view" [1]). In this proposed architecture the attention operation associates points lying on the same manifold most strongly (as per the inner-product findings above), and the MLPs are responsible for computing the signatures of each point (i.e. the separate concepts). Applied across multiple layers this amounts to a hierarchical computation of signature, and thus concepts, associated with the inputs.

The authors make their theoretical framework somewhat more concrete by providing examples of signatures corresponding to circles, moving objects and rotation-invariant point clouds.

[1] Elhage et al. 2021 - A Mathematical Framework for Transformer Circuits

**Strengths:**

The mathematical treatment of the proposed framework for concept discovery is thorough, and the examples aid in understanding. Additionally, the proposed architecture appears promising from a theoretical perspective. To the best of our knowledge, formalising concept discovery as membership on learned polynomial manifolds is novel.

**Weaknesses:**

There are two primary weaknesses of the paper, though these may easily reflect a lack of understanding on the part of the reviewer:
1) It is very dense, though this is somewhat expected of a theoretical paper and,
2) It is unclear whether this framework is particularly useful, either for understanding concept acquisition on currently-used transformer architectures, or for learning in their proposed architecture. For example, there are good reasons to believe that Capsule-Nets should learn to hierarchically decompose complex objects into re-usable parts, but most of the existing literature on Part-Whole hierarchies has encountered great difficulties in being applied to non-toy datasets when training with e.g. gradient descent. As such, the lack of experiments makes it unclear that a reader is well-served to work through this theoretical paper, as it is not immediately obvious that theoretical guarantees around inner-product proximity and signature computation will reflect the learning of disentangled concept manifolds in practice.

Nitpick:
1. The first sentence of section 3.1 is missing "*can be used* to group points"

**Questions:**

We have one high-level question regarding the practical applicability of this framework: Which constraints, if any, does the assumption of a polynomial manifold impose on the form of the input vectors which this framework could model? In particular, are there reasons to expect the assumption of linear manifolds associated with concepts to hold in practical settings, where e.g. the inputs, $x_{t}$, might be features from a complex neural network?

---

> ### Author Response · Authors · 2023-11-15
> **Answers questions about representations**
>
> We thank the reviewer for their time and effort in providing feedback on our work. We agree with the reviewer that the present paper lacks experimental work. The paper aimed to take a first-principles approach to the challenging problem of formalizing the recognition of concepts and deduce the implications of such a formalism. While there are several empirical investigations of such questions for existing architectures and methods, the formalism of concepts is quite poorly understood, and it would help to develop theoretical formalism first without being bogged down by the noise in data, variations in architectures, training methods, and datasets. Designing extensive experiments and developing the framework for real-world datasets will be a substantial body of work by itself, just as developing the formalism here was.
>
> Regarding the question about polynomial manifolds, the reviewer is right that this is an assumption that constraints on the inputs. However, the embedding mapping is meant to capture them and allows for wider applicability.
>
> Regarding the question about polynomial manifolds, the reviewer is right that we are constraining vector representations of different instances of a concept to change smoothly and lie in a manifold however, we believe that even if this is not true in the input layer as one computes signatures up the layers hierarchically one gets smoother representations . Although we have argued this  concretely only for simple concepts such as circles,  rectangles, and stick figures we believe this should be true more generally.
>
> We hope that our response resolves the concerns of the reviewer and we are happy to discuss more to clarify any further concerns.

---

> > ### Comment · Reviewer_G7CX · 2023-11-22
> > **Response to clarifications and changes**
> >
> > Thank you for your detailed response, and for the numerous changes to the paper which have improved its clarity, though it remains dense.
> >
> > I am willing to agree with the intuition that a learned embedding should be expressive enough to warrant the assumption of inputs lying on a polynomial manifold, and that in a deep transformer, subsequent layers may learn smooth representations/deformation thereof.
> >
> > My key concern remains that there are many practical issues which may limit the utility of the proposed theory, and which are not directly addressed by the theory. These include:
> > * Whether clear, and strongly disentangled signatures / concepts would be learned by the proposed model under realistic training conditions
> > * If so, whether the hierarchy of learned concepts would correspond to something more meaningful than the entangled, but hierarhical, features learned by almost any deep neural network
> > * Whether these concepts would be sufficiently sharply disentangled that they could be meaningfully "manipulated" (as per the title)
> >
> > Nonetheless, I appreciate that a succinct theory which can be straightforwardly falsified in future work may warrant publication in of itself. As such I will not change my scores, but have increased my confidence based upon your clarifications.

---

### Official Review · Reviewer_bdWm · 2023-11-09

**Soundness:** 3 good
**Presentation:** 3 good
**Contribution:** 2 fair
**Rating:** 6
**Confidence:** 2

**Summary:**

The authors propose to formalize concepts to be a shared set of properties defined by a polynomial manifold. Such manifold can be identified by the null-space of moment statistics of points in the manifold, which makes these statistics serve as the signature of the manifold, or concept. The strengths of such signatures are that the structure of concepts can be reflected in the signature of concepts and the lower-level signature can be used to derive higher-level signatures. The authors also analyze the transformer network on how it captures the concepts defined in this paper.

**Strengths:**

1. The authors provide a novel formalism of "concepts", which facilitates further studies on "concepts";
2. The authors provide proof for every statement they propose. They also provide examples to illustrate some abstract ideas;

**Weaknesses:**

Although the analyses and demonstrations are comprehensive, the authors do not provide any experiments to support their theories or how their theories can be applied to existing models. I suggest authors may conduct a few experiments to show the applicability of their model.

**Questions:**

See my suggestions above.

---

> ### Author Response · Authors · 2023-11-15
> **Concerns about experiments**
>
> We thank the reviewer for their time and effort in providing feedback on our work. We agree with the reviewer that the present paper lacks experimental work. The paper aimed to take a first-principles approach to the challenging problem of formalizing the recognition of concepts and deduce the implications of such a formalism. While there are several empirical investigations of such questions for existing architectures and methods, the formalism of concepts is quite poorly understood, and it would help to develop theoretical formalism first without being bogged down by the noise in data, variations in architectures, training methods, and datasets. Designing extensive experiments and developing the framework for real-world datasets will be a substantial body of work by itself, just as developing the formalism here was.
>
> We hope our response resolves the reviewer's concerns, and we are happy to discuss further to clarify any further concerns.

---

> > ### Comment · Reviewer_bdWm · 2023-11-22
> >
> > Thanks for your response. I agree that this problem is a very important direction for many application domains. However, I still feel that experiments on real-life datasets or toy examples are required to justify the proposed framework.

---

### Author Response · Authors · 2023-11-23
**Experiments on synthetic data**

We thank all reviewers for their reviews and we are excited that the reviewers found the formalism of recognition of concepts novel and sound.

We have updated the paper to include experiments on synthetic datasets in Appendix E to show the effectiveness of our proposed method and architecture in recognizing and separating the concepts in a modular way.

We want to reiterate that the paper aims to develop a theoretical framework for the problem of formalizing concept recognition, a topic that remains poorly understood even though there are several empirical works about it. Designing extensive experiments and developing a framework for real-world datasets is an interesting future direction of this work but it is outside the scope of a single project that aims to develop the theoretical framework for the challenging problem.

We hope our response and the new draft resolve the reviewer's concerns.

---

### Meta-Review · Area_Chair_ZMPg · 2023-12-04

**Metareview:**

This paper introduces a formalism on learning of concepts and claims that it can be achieved via generating a signature of each concept, which can be then used to discover structure and find common themes in a number of related concepts. The paper thoroughly formalizes the setting that is considered and discusses implications and possible architectures that could leverage the proposal.

While reviewers consistently recognized the importance of making progress towards learning concepts, they unanimously demonstrated concerns regarding the scope of the work being not broad enough since there's not enough empirical evidence of the usefulness of the proposed framework or to what extent naturally occurring data satisfy the underlying assumptions. There are also issues with presentation and the submitted manuscript does not fully follow the conference's template. I would also add that, as pointed out by the authors, empirical methods do exist under a similar setting, and the paper should include a section discussing connections with related work. Given these limitations, we conclude that this paper is not ready for acceptance in its current form.

**Justification For Why Not Higher Score:**

While the paper is quite interesting, there are multiple issues that should be addressed before it being ready for publication as discussed in the latter portion of the meta-review.

**Justification For Why Not Lower Score:**

N/A

---

### Decision · Program_Chairs · 2024-01-16

Reject